# Does the use of statins alter the risk of rheumatoid arthritis? A systematic review and meta-analysis

**Xinhong Pan, Xiaobing Yang, Peiying Ma, Li Qin** [ORCID] *

Department of Rheumatology and Immunology, Huzhou Third Municipal Hospital, The Affiliated Hospital of Huzhou University, Huzhou City, Zhejiang Province, China

* qinliqinli2023@163.com

## Abstract

### Objective

Statins have anti-inflammatory and immune-modulatory effects which could alter the risk of rheumatoid arthritis (RA). We reviewed published literature and conducted a meta-analysis to examine if statins have an impact on the risk of RA.

### Methods

Case-control studies, cohort studies, or randomized controlled trials (RCT) published on the PubMed, Scopus, and EMBASE databases up to 30th October 2023 were searched. The association between statin use and risk of RA was pooled in a random-effects meta-analysis.

### Results

Nine studies (four cohort, four case-control, and one RCT) were included. Overall, the analysis failed to note an association between the use of statins and the risk of RA with the pooled OR being 0.93 (95% CI 0.82, 1.06). High heterogeneity was noted with $I^2 = 75\%$. Results were consistent across study types with no association noted between prior statin use and risk of RA in case-control studies (OR: 0.88 95% CI: 0.69, 1.13), cohort studies (OR: 1.01 95% CI: 0.92, 1.10), and the lone RCT (OR: 1.40 95% CI: 0.50, 3.92).

### Conclusion

Current literature shows that there is no association between the use of statins and the risk of RA. Further rigorous studies taking into account patient factors, duration of statin exposure, and other confounders are needed to generate better evidence.

## Introduction

Rheumatoid arthritis (RA) is a common intractable auto-immune condition which mostly affects the joints. It is often rapid in onset resulting in significant functional disability and early

**Data Availability Statement:** All relevant data are within the manuscript and its Supporting information files (PubMed, Scopus, and EMBASE databases).

**Funding:** The author(s) received no specific funding for this work.

**Competing interests:** The authors have declared that no competing interests exist.

disease-related mortality [1]. Global prevalence has been reported to be around 0.46% ranging from 0.39 to 0.54% [2]. While RA can develop at any age, the risk increases in older adults with a preponderance in those aged above 50 years [3]. Such elderly patients frequently require through nursing care to alleviate joint pain, promote joint mobility, and education on self-care strategies. As the world population ages, the prevalence of RA is expected to rise. Hence, disease-modifying drugs must be identified to lower the burden of RA in the future.

Statins, or 3-hydroxy-3-methylglutaryl coenzyme A (HMG-CoA) reductase inhibitors are amongst the most commonly prescribed drugs for hyperlipidemia and reducing the risk of cardiovascular disease [4]. Given the high burden of obesity and cardiovascular risk, statins have become one of the highest prescribed drugs in the USA [5]. In addition to the lipid-lowering function, other pleiotropic effects have also been recognized [6]. They can alter a number of non-lipid-related cell signaling pathways including those involved in inflammatory responses [7]. Randomized trials have provided evidence on the anti-inflammatory effect of statins in the general population and those with chronic inflammatory conditions like RA [8, 9]. Statins have shown to be successful in ameliorating RA activity by downregulating inflammatory factors and reducing joint [8, 10, 11]. On the other hand, research also shows that long-term statin use may induce autoimmune reactions leading to the development of rheumatic diseases like systemic lupus erythematosus, dermatomyositis, and polymyositis [12]. Such conflicting evidence has also been found for long-term statin use and the risk of RA. Some studies have found a protective role of statins on RA [13] while others have noted an increased risk [14]. Therefore, we performed the current systematic review and meta-analysis to compile data from published literature and provide high-quality evidence on the role of statins in altering the risk of RA.

## Material and methods

### Search strategy and inclusion criteria

The question to be answered by this review was: "Does the use of statins alter the risk of incident RA?". For this, a systematic literature search of PubMed, Scopus, and EMBASE databases pertaining to the clinical question was concluded on 30[th] October 2023. The reviewers were guided by the PRISMA guidelines [15] for the design, execution, and presentation of the review and meta-analysis. Pre-registration was done on PROSPERO (CRD42023474177). English language studies were identified using a common search string utilized for the databases. This was: ((((((Simvastatin) OR (Pravastatin)) OR (Atorvastatin)) OR (Fluvastatin)) OR (Rosuvastatin)) OR (statin)) AND (rheumatoid arthritis). Two reviewers independently screened the results after electronic deduplication to determine if the article met the inclusion criteria. The inclusion criteria were: 1) Case-control studies, cohort studies, or randomized controlled trials (RCT) published as full-length articles or abstracts. 2) Examined the association between statin use and risk of RA. 3) Exposure was the use of statins and the outcome was RA. 4) Reported an effect ratio for the association with 95% confidence intervals (CI) or reported sufficient data for the same to be calculated. The reviewers excluded cross-sectional studies, editorials, and unpublished data. Studies on arthritis in general and not specifically on RA were ineligible for inclusion.

The full text of relevant articles was further independently reviewed by two reviewers, and differences were discussed with a third reviewer to reach a consensus, and the references of selected articles were checked to discover other relevant papers. As this is a review of previously published articles, participants' informed consent and ethical approval are not needed.

## Extracted data and study quality

Extracted data from studies included author details, year of publication, study type, study database and period, sample size, method of identification of statin exposure and RA, number exposed to statins, number of cases with RA, follow-up, and effect ratio. Any subgroup analysis conducted by the studies was also reproduced for completeness of evidence. Two reviewers were involved in data collection and all data was cross-checked again by the primary article in case of discrepancies in data collection.

Observational studies were assessed for their methodological quality by two reviewers using the Newcastle Ottawa Scale (NOS) [16]. Points were awarded for the representativeness of the study cohort, comparability of groups, and measurement of outcomes with each receiving a maximum of four, two and three points respectively. RCTs were examined for risk of bias based on the Cochrane Collaboration risk of bias-2 tool [17].

## Statistical analysis

Quantitative synthesis was carried out by "Review Manager" (RevMan, version 5.3; Nordic Cochrane Centre (Cochrane Collaboration), Copenhagen, Denmark; 2014). The effect ratios were initially extracted in tabular form. Given the small risk of RA with the exposure to statin, odds ratios, hazard ratios, and risk ratios were considered to have minimal differences and were combined in a single meta-analysis. Forest plots were produced in the software by using the random-effect meta-analysis model. The generic inverse variance function was used to combine log-transformed values of the effect ratios. Adjusted data was used when available. Between studies, heterogeneity was examined by $I^2$ statistic with a value of >50% meaning substantial heterogeneity. Subgroup analysis was done based on study type. Funnel plots were checked for publication bias. Leave-one-out analysis was conducted in the software itself to check for the robustness of the results.

## Results

On completion of the database search, 1150 articles were retrieved. Following the removal of duplicates, 528 underwent further evaluation. 508 were not deemed to be relevant to the review and hence excluded. 20 studies were selected for further review and nine made it to the meta-analysis [13, 14, 18–24] (Fig 1).

Details extracted from the studies are presented in Table 1. There were four cohort studies, four case-control studies and one prematurely stopped RCT. Of the cohort studies, two were prospective and two were retrospective. Two cohort studies were matched for baseline variables. All case-control and cohort studies used a local or national registry for data extraction. The only RCT could include just 62 patients owing to low recruitment. Amongst the non-RCT, the sample size ranged from 1565 to 2121786 participants. Prescription records were used to identify statin users in all studies. Except for the RCT, all studies used medical records to ascertain RA diagnosis. Duration of follow-up was not reported in the majority of studies. The NOS score for all three domains for observational studies is presented in Table 1. The RCT was found to have low risk of bias for randomization process, deviation from intended intervention, measurement of outcomes, and selection of reported results. However, there was high risk of bias due to missing outcome data owing to high number of drop-outs. Hence, overall the risk of bias for the trial was high.

Meta-analysis of all studies with subgroup analysis based on study type is shown in Fig 2.

Overall, the analysis failed to note an association between the use of statins and risk of RA with the pooled OR being 0.93 (95% CI 0.82, 1.06). High heterogeneity was noted with $I^2$ = 75%. Results were consistent across study types with no association noted between prior statin

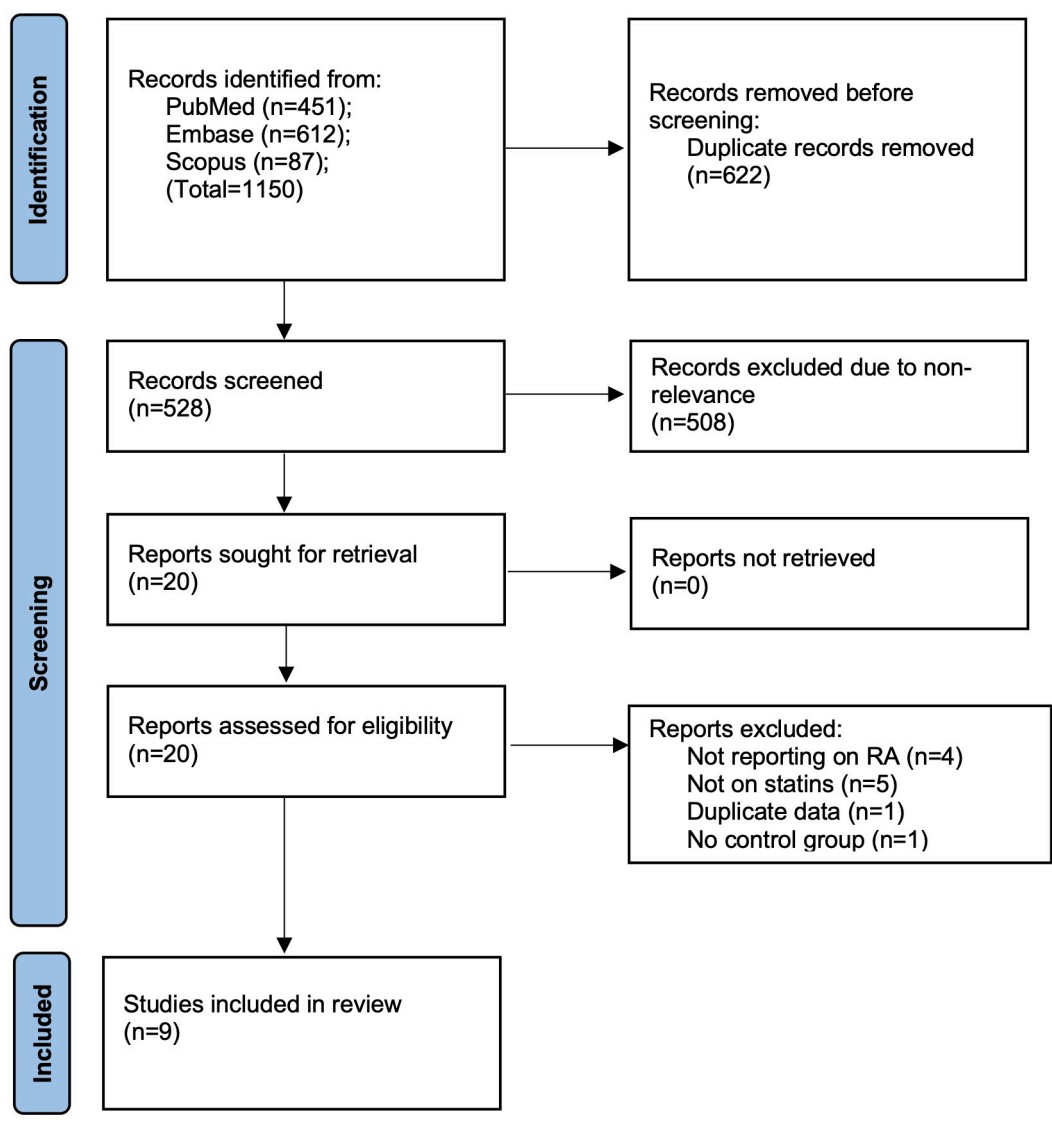

**Fig 1. Flow-chart of study selection.**

use and risk of RA in case-control studies (OR: 0.88 95% CI: 0.69, 1.13), cohort studies (OR: 1.01 95% CI: 0.92, 1.10), and the lone RCT (OR: 1.40 95% CI: 0.50, 3.92). There was no publication bias on the funnel plot (Fig 3). There was no change in the significance of the results on exclusion of any study on sensitivity analysis.

The included studies reported subgroup analysis based on different variables. Relevant details are presented in Table 2. Three studies reported the association based on the type of statin. None of the studies reported any significant difference across types of statins, except for De Jong et al [14] wherein the use of atorvastatin was associated with an increased risk of RA. Three studies reported data on past statin use only to find no significant association. Three studies reported data based on gender. Two of them reported no association between statin use and risk of RA in both men and women. The study of Kwon et al [13] found a reduced risk of RA with the use of statins in both men and women. Two studies reported subgroup analysis based on the duration of statin use (> or ≤ 1 year). Only Peterson et al [18] found that the use

**Table 1. The characteristics of each study.**

| References | Type | Database | Study period | Sample size | Identification of statin exposure | Identification of RA | % exposed to statins | % with RA | Follow-up | NOS | Effect size |
|---|---|---|---|---|---|---|---|---|---|---|---|
| Jick (2009) | Case-control | General practice research database | 1992–2001 | 1565 | Prescription records | ICD and Read codes | 15 | 20 | NR | S****<br>C**<br>E* | OR 0.59 (0.37, 0.96) |
| Smeeth (2009) | Cohort, R | The Health improvement network database | 1995–2006 | 729529 | Prescription records | Medical records | 17.7 | 0.34 | 4.4 years | S**<br>C**<br>O* | HR: 0.93 (0.73, 1.18) |
| Hippisley-Cox (2010) | Cohort, P | Q research database | 2002–2008 | 2121786 | Prescription records | Read codes | 10.7 | 0.2 | NR | S***<br>C**<br>O** | HR: 1.02 (0.88, 1.17) |
| Schmidt (2013) | Matched cohort, R | San Antonio military medical community | 2003–2010 | 13912 (matched cohort from a sample of 46488) | Prescription records | ICD codes | 50 | 1.6 | NR | S****<br>C**<br>O** | OR: 0.85 (0.65, 1.11) |
| De Jong (2012) | Case-control | Netherlands Information Network of General Practice database | 2001–2006 | 2877 | Prescription records | ICD codes | 8 | 17.6 | NR | S****<br>C**<br>E* | OR: 1.71 (1.16, 2.53) |
| De Jong (2018) | Matched cohort, P | Clinical Practice Research Datalink | 1995–2009 | 1023240 | Prescription records | Read codes | 50 | 0.1 | 3± 2.5 years | S****<br>C**<br>O** | HR: 1.06 (0.93, 1.22) |
| Boheemen (2021) | RCT | STAPRA trial | 2015–2019 | 62 | Prescribed during the trial | 2010 ACR/ EULAR classification criteria | 50 | 24 | NR | - | HR: 1.40 (0.50, 3.95) |
| Peterson (2021) | Case-control | OptumLabs Data Warehouse | 2010–2019 | 32726 | Prescription records | ICD codes | 0.16 | 50 | NR | S****<br>C**<br>E* | OR: 0.87 (0.81, 0.93) |
| Kwon (2022) [13] | Case-control | Korean National Health Insurance Service | 2002–2015 | 4998 | Prescription records | ICD codes | 10.87 | 50 | NR | S****<br>C**<br>E* | OR: 0.73 (0.63, 0.85) |

Abbreviation: P, prospective; R, retrospective; ICD, international classification of diseases; RA, rheumatoid arthritis; RCT, randomized controlled trial; NR, not reported; S, selection; C, comparability; O, outcome assessment; NOS, Newcastle Ottawa scale; OR, Odds ratio; HR, hazard ratio

of statin for >1 year was protective of RA. Given the heterogeneity of subgroup variables and the limited number of studies for each subgroup, a separate meta-analysis for such data was not conducted.

## Discussion

In the past decade, there has been growing interest in the immunomodulatory role of statins and the possibility of altering the risk of autoimmune and inflammatory diseases. Ungaro et al [25] in a large observational study using USA healthcare records reported that statin use was associated with reduced risk of new-onset inflammatory bowel disease including both ulcerative colitis and Crohn's disease. They noted that the protective effect was noted across all statins with the elderly being most benefited. Lin et al in a recently published article analyzing

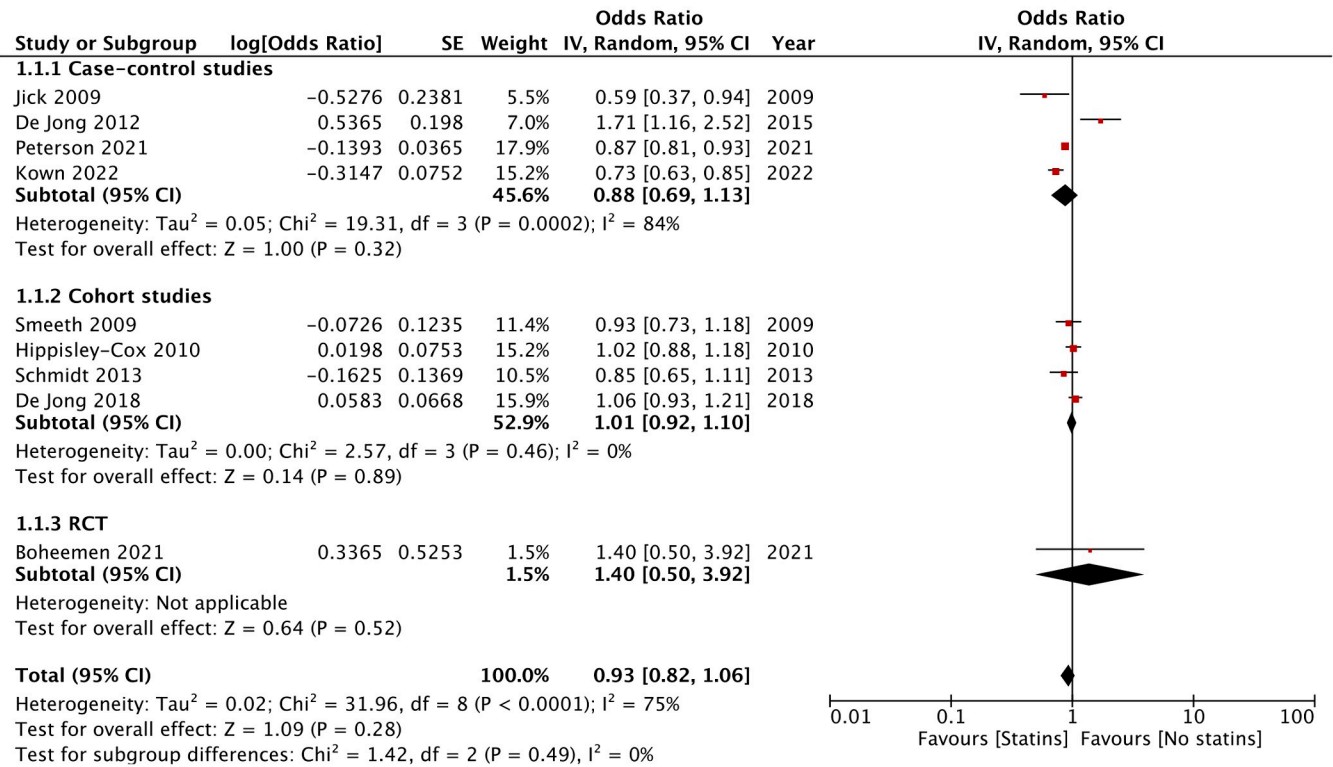

**Fig 2. Meta-analysis of the association between statin use and risk of RA based on study type.**

data from the Taiwan healthcare database have shown that those receiving large cumulative doses and prolonged therapy of statins had a lower risk of gout. Almramhi et al [26] in a Mendelian randomization found that the pleiotropic effects of statins may have a role in reducing the risk of multiple sclerosis independent of the cholesterol pathway. Contrastingly, there is also evidence suggesting a harmful effect of statins. Noel et al reviewed 28 cases of statin-induced autoimmune diseases and found lupus erythematosus being the common disease followed by dermatomyositis and polymyositis. There have also been numerous reports of statin-induced autoimmune myopathy mostly seen in male and elderly individuals [27].

A similar contrasting association has been reported for statin use and risk of RA, as noted in included studies. In this context, the current systematic review and pooled analysis provide important cumulative evidence. On pooled analysis of all nine studies, we noted that prior use of statins was not associated with any change in the risk of RA. Due to the different study types included, a subgroup analysis was also conducted. Combined analysis of case-control studies also demonstrated no association between statin use and risk of RA. However, out of the four case-control studies, three noted a reduced risk of RA with statin while one noted a protective effect. Also, there was high heterogeneity noted in this meta-analysis ($I^2 = 84\%$). Similar results were noted for the meta-analysis of cohort studies but with more consistency amongst the four studies with none of them reporting a positive or negative association between statin use and risk of RA. The interstudy heterogeneity in this meta-analysis was nil thereby providing more robust evidence.

Without a doubt, the best evidence of the association between statins and RA can be generated only by rigorous RCTs. However, prevention trials are often difficult to conduct due to problems in patient recruitment [28]. Furthermore, there are other issues like the need for

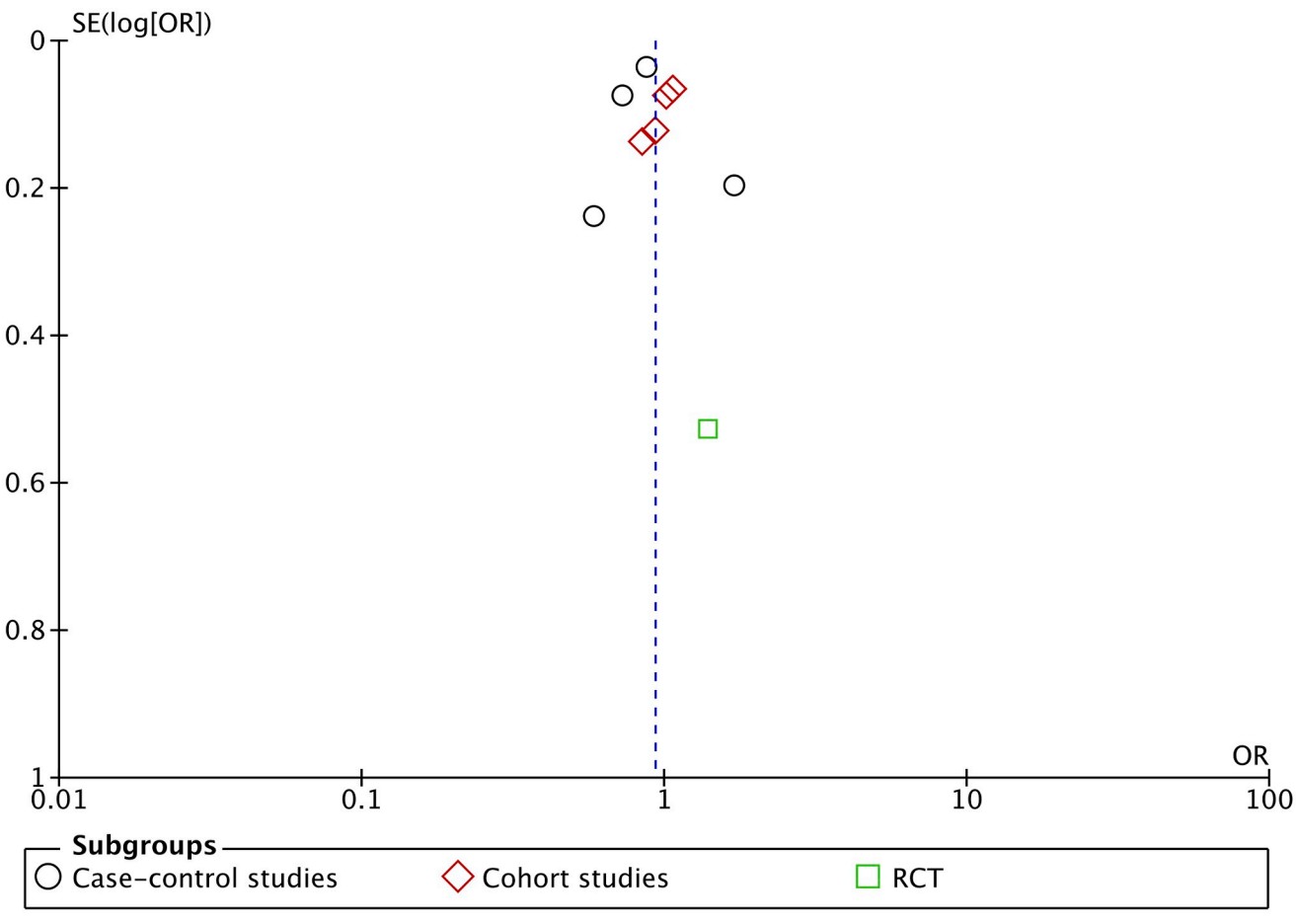

**Fig 3. Funnel plot of the meta-analysis.**

long-term follow-up to monitor the outcome. The only RCT included in the meta-analysis also suffered from recruitment problems and had to be stopped prematurely. Only a small number of participants could be included in the trial with a large number of patients being lost to follow-up. Given such difficulties, clinicians will have to rely on rigorous cohort studies to ascertain the relationship between statins and RA.

Previously, Myasoedova et al [29] have also conducted a meta-analysis on the same topic. While our results were similar to the previous review, there are important differences. The current review is an updated review including nine studies, three more than the previous review. Myasoedova et al [29] could conduct a meta-analysis of only four cohort studies. However, in our review, a meta-analysis was conducted for different study types including a descriptive analysis of various subgroups to further understand the role of statins in the risk of RA. We could not pool data of different subgroups owing to major differences amongst included studies. However, the descriptive analysis indicates that in most subgroups, there was no association between statin use and the risk of RA. It was only in the study of Kwon et al that reduced risk of RA was noted with statin use across multiple subgroups which was consistent with the overall results of the study.

Several different possible mechanisms have been put forward suggesting a harmful or protective effect of statins on RA. Statins have been shown to lower T helper cells (Th1)/Th2 and

**Table 2. Subgroup analysis reported by the included studies.**

| References | Variable | Subgroup | Effect size |
|---|---|---|---|
| **Jick (2009)** | Type of statin | Simvastatin | OR: 0.74 (0.45, 1.23) |
| | | Pravastatin | OR: 0.41 (0.15, 1.12) |
| | | Atorvastatin | OR: 0.42 (0.12, 1.49) |
| | Timing | Past statin | OR: 0.79 (0.21, 2.96) |
| **Hippisley-Cox (2010)** [22] | Gender | Women | HR: 0.95 (0.86, 1.05) |
| | | Men | HR: 1.10 (0.97, 1.24) |
| | Type of statin | Simvastatin | HR: 1.03 (0.89, 1.020) |
| | | Pravastatin | HR: 0.93 (0.64, 1.33) |
| | | Atorvastatin | HR: 1.01 (0.87, 1.17) |
| | | Fluvastatin | HR: 1.03 (0.62, 1.71) |
| | | Rosuvastatin | HR: 0.80 (0.47, 1.38) |
| **De Jong (2012)** | Number of prescriptions | 1–4 | OR:2.25 (1.27–3.98) |
| | | 5–8 | OR: 1.62 (0.86–3.03) |
| | | ≥9 | OR: 1.59 (0.90–2.81) |
| | Cumulative duration | 1–250 days | OR: 1.59 (0.85–2.95) |
| | | 251–600 days | OR: 1.50 (0.85–2.64) |
| | | ≥601 days | OR: 1.38 (0.85–2.23) |
| | Cumulative daily dose | 1–300 | OR: 1.67 (0.92–3.04) |
| | | 301–900 | OR: 1.32 (0.74–2.35) |
| | | ≥901 | OR: 1.44 (0.90–2.33) |
| | Type of statin | Simvastatin | OR: 1.53 (0.89–2.62) |
| | | Pravastatin | OR: 1.66 (0.78–3.53) |
| | | Fluvastatin | OR: 2.65 (0.72–9.72) |
| | | Atorvastatin | OR: 2.35 (1.29–4.29) |
| | | Rosuvastatin | OR: 1.69 (0.38–7.61) |
| **De Jong (2018)** | Timing | Past users | HR: 1.18 (0.88 to 1.57) |
| | | ≤1 year | HR: 1.27 (1.00 to 1.61) |
| | | >1 year | HR: 0.98 (0.80 to 1.19) |
| | Gender | Women | HR: 1.17 (0.93 to 1.47) |
| | | Men | HR: 0.89 (0.66 to 1.21) |
| | Age | 40–50 | HR: 1.29 (0.63 to 2.65) |
| | | 51–60 | HR: 0.94 (0.66 to 1.33) |
| | | 61–80 | HR: 1.01 (0.79 to 1.28) |
| | | >80 | HR: 0.93 (0.43 to 2.02) |
| **Peterson (2021)** | Timing | Past users | OR: 1.03 (0.96–1.10) |
| | | ≤1 year | OR: 1.04 (0.94–1.15) |
| | | >1 year | OR: 0.91 (0.86–0.97) |
| **Kwon (2022)** | Age | <60 | OR: 0.71 (0.54–0.92) |
| | | >60 | OR: 0.77 (0.64–0.92) |
| | Gender | Women | OR: 0.76 (0.64–0.91) |
| | | Men | OR: 0.62 (0.47–0.82) |
| | Weight | Normal weight | OR: 0.57 (0.43–0.77) |
| | | Overweight | OR: 0.65 (0.50–0.86) |
| | | Obese | OR: 0.93 (0.75–1.16) |
| | Smoking | Smoker | OR: 0.63 (0.45–0.88) |
| | | Non-smoker | OR: 0.75 (0.64–0.88) |
| | Dyslipidemia | No history | OR: 1.18 (0.75–1.86) |
| | | Positive history | OR: 0.72 (0.61–0.84) |

Abbreviation: OR, Odds ratio; HR, hazard ratio

CD4/CD8 ratio resulting in an anti-rheumatic effect with improvement in symptoms in RA [30]. These drugs also lead to reduced production of IL-6 and IL-8 along with IL-1 stimulated fibroblast-like-synoviocytes causing a reduced inflammatory response [31]. Statins also inhibit 3-hydroxy-3-methyl-glutaryl-coenzyme A reductase causing depletion of l-mevalonate pathway downstream metabolites leading to reduced severity of the autoimmune diseases [32]. Contrastingly, there is evidence that statins increase immune cell inflammatory function. Lipopolysaccharide simulated macrophages treated with statins have shown increased inflammasome signalling with heightened IL-1β release [33]. Natural killer cells treated with statins and IL-2 have shown increased production of cytokines with cytotoxic effects on tumour cells [34]. Fluvastatin has been shown to simulate IL-33-mediated mast cell activation increasing tumor necrosis factor and IL-6 production [35]. Indeed, the variation in preclinical studies taken together with conflicting clinical evidence on the association between statins and RA indicates that individual patient characteristics, the type, duration, and intensity of statin use could be important factors affecting the relationship. At this point, the evidence does not support the use of statins for the prevention of RA and there is no need for change in clinical guidelines, however, there is a need for further robust research to assess the relationship. It is recommended that nursing personnel involved in care of patients on long-term statins should be well-versed with the pleiotropic effects of statins and the current conflicting evidence. They should monitor patients for early signs of RA as some of the studies have shown increased risk with statins.

There are limitations to this meta-analysis. A mix of case-control and cohort studies predominated the review. Only one RCT was available which too of a very small sample size. The heterogeneity in the meta-analysis was high suggesting caution in the interpretation of the results. The heterogeneity could be due to several reasons like patient characteristics, comorbidities, timing of intervention, dose and duration of statin therapy, follow-up, etc. Limited studies and a lack of baseline data precluded a subgroup analysis for such factors. Additionally, the follow-up duration was not consistently reported across studies. It is currently unknown how long it takes for RA to develop. Some studies may not have adequately followed up the exposed cohort to identify the development of RA. Another factor to consider is that the included studies used different ratios, OR or hazard ratio or risk ratio to report the association between statin use and RA. Due to low prevalence of RA and limited number of studies in literature, we were forced to combine these ratios in a single meta-analysis. Given these ratios are not exactly the same, there could be some bias in our results. Lastly, there was a predominance of Western data in the meta-analysis limiting the generalizability of the results.

## Conclusions

Current literature shows that there is no association between the use of statins and the risk of RA. Further rigorous studies taking into account patient factors, duration of statin exposure, and other confounders are needed to generate better evidence.

## Supporting information

**S1 Checklist. PRISMA 2020 checklist.**
(DOCX)

## Author Contributions

**Conceptualization:** Xinhong Pan, Xiaobing Yang, Peiying Ma, Li Qin.

**Data curation:** Xinhong Pan, Xiaobing Yang, Peiying Ma, Li Qin.

**Formal analysis:** Xinhong Pan, Xiaobing Yang, Peiying Ma, Li Qin.

**Investigation:** Xiaobing Yang.

**Methodology:** Xinhong Pan, Xiaobing Yang, Peiying Ma, Li Qin.

**Software:** Peiying Ma.

**Supervision:** Peiying Ma, Li Qin.

**Validation:** Xinhong Pan, Xiaobing Yang, Peiying Ma, Li Qin.

**Visualization:** Xinhong Pan, Xiaobing Yang, Peiying Ma, Li Qin.

**Writing – original draft:** Xinhong Pan, Li Qin.

**Writing – review & editing:** Li Qin.

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
