## [Decision Letter · Decision Letter 0]

14 Jun 2024

PONE-D-24-17747Does the use of statins alter the risk of rheumatoid arthritis? A systematic review and meta-analysisPLOS ONE

Dear Dr. Qin,

Thank you for submitting your manuscript to PLOS ONE. After careful consideration, we feel that it has merit but does not fully meet PLOS ONE’s publication criteria as it currently stands. Therefore, we invite you to submit a revised version of the manuscript that addresses the points raised during the review process.

We look forward to receiving your revised manuscript.

Kind regards,

Elena Olmastroni

Academic Editor

PLOS ONE

Journal Requirements:

4. We note that you have referenced (unpublished) on page 4,  which has currently not yet been accepted for publication. Please remove this from your References and amend this to state in the body of your manuscript: (ie “Bewick et al. [Unpublished]”) as detailed online in our guide for authors

Reviewers' comments:

Reviewer's Responses to Questions

**Comments to the Author**

1. Is the manuscript technically sound, and do the data support the conclusions?

Reviewer #1: Yes

Reviewer #2: Partly

2. Has the statistical analysis been performed appropriately and rigorously? 

Reviewer #1: Yes

Reviewer #2: No

3. Have the authors made all data underlying the findings in their manuscript fully available?

Reviewer #1: Yes

Reviewer #2: Yes

4. Is the manuscript presented in an intelligible fashion and written in standard English?

Reviewer #1: Yes

Reviewer #2: Yes

5. Review Comments to the Author

Reviewer #1: This is a systematic review and meta-analysis on the impact of statins on the risk of rheumatoid arthritis. Although simple, was very well conducted and discussed considering scientific knowledge. I congratulate the authors for their manuscript.

Reviewer #2: The authors conducted a systematic review and meta-analysis evaluating the potential link between statin use and the risk of developing rheumatoid arthritis. However, as mentioned by the authors, this topic had been clearly addressed by the previous published meta-analysis (doi: 10.1016/j.semarthrit.2020.03.008). There are 2 studies (Tascilar [2016] and Chodick [2010]) had been included in the previous one but not the current submission. Please clarify the reasons for not including those two studies. In addition, there are some issues regarding the methodology used in this submission.

1. The authors included both randomized and non-randomized studies, which is not recommended by the Cochrane Handbook due to the risk of bias.

2. The authors assessed the quality of all included studies by using the Newcastle Ottawa Scale (NOS), but it is a tool used only for the non-randomized studies. For randomized controlled trials, the Cochrane Risk of Bias (RoB) is recommended.

3. The authors mentioned that “Given the small risk of RA with the exposure to statin, odds ratios, hazard ratios, and risk ratios were considered to have minimal differences and were combined in a single meta-analysis.”, which I think is incorrect. I have checked the included studies and found even both odds ratios (ORs) reported in the articles (for example, Schmidt [2013] and de Jong [2012]), there were significant differences in the way they calculated the ORs. This could also be one of reasons for the high heterogeneity. I would suggest using raw data (number of events) and pooled with the same ratio (OR or RR).

4. As several of the included studies mentioned that intervention durations of less than or more than 1 year may affect the results. Subgroup analysis based on intervention duration is needed.

5. It is also important to conduct the leave-one-out analysis to determine whether the results could have been influenced by a single study.

6. PLOS authors have the option to publish the peer review history of their article (what does this mean?). If published, this will include your full peer review and any attached files.

Reviewer #1: No

Reviewer #2: No

---

## [Author Response · Author response to Decision Letter 0]

19 Jun 2024

Journal Requirements:

Response: Done

Response: The article contains all data analyzed in the meta-analysis

Response: Done

4. We note that you have referenced (unpublished) on page 4, which has currently not yet been accepted for publication. Please remove this from your References and amend this to state in the body of your manuscript: (ie “Bewick et al. [Unpublished]”) as detailed online in our guide for authors

Response: We have written on page 4 that: “The reviewers excluded cross-sectional studies, editorials, and unpublished data.” This means we have not included any unpublished data in the manuscript.

Comments to the Author

Reviewer #1: This is a systematic review and meta-analysis on the impact of statins on the risk of rheumatoid arthritis. Although simple, was very well conducted and discussed considering scientific knowledge. I congratulate the authors for their manuscript.

Response: Thank you for your comments.

Reviewer #2: The authors conducted a systematic review and meta-analysis evaluating the potential link between statin use and the risk of developing rheumatoid arthritis. However, as mentioned by the authors, this topic had been clearly addressed by the previous published meta-analysis (doi: 10.1016/j.semarthrit.2020.03.008). There are 2 studies (Tascilar [2016] and Chodick [2010]) had been included in the previous one but not the current submission. Please clarify the reasons for not including those two studies. In addition, there are some issues regarding the methodology used in this submission.

Response: Thank you for reviewing our manuscript and providing constructive comments. We agree with the reviewer that 2 studies (Tascilar [2016] and Chodick [2010]) included in the previous review were not included in our review, but for important reasons. The study of Tascilar 2016 is in complete overlap with de Jong 2018. Since de Jong 2018 had a longer study duration and larger sample size, we excluded Tascilar 2016. The study of Chodick et al assessed the risk of bias with “persistence of statins”. In their cohort all patients were users of statins and they divided their study population into multiple groups based on number of days of covered with statins (<20%, 20-29%,…>80%). There was no control group of “no statins” in their study. Hence, this study was not included. We have edited Figure 1 to update the reasons for exclusion as these were not reflected earlier.

1. The authors included both randomized and non-randomized studies, which is not recommended by the Cochrane Handbook due to the risk of bias.

Response: We agree with the point of view of the reviewer. Hence, we have already conducted a subgroup analysis based on study designs. Our aim was to present comprehensive evidence on the association between statin use and risk of RA. This would have been incomplete without presenting the results of the only RCT conducted on this topic. 

2. The authors assessed the quality of all included studies by using the Newcastle Ottawa Scale (NOS), but it is a tool used only for the non-randomized studies. For randomized controlled trials, the Cochrane Risk of Bias (RoB) is recommended.

Response: This error is now corrected. We have now used the Cochrane Risk of Bias tool. Since there was only one RCT, results are presented in the manuscript itself with the following lines: “The RCT was found to have low risk of bias for randomization process, deviation from intended intervention, measurement of outcomes, and selection of reported results. However, there was high risk of bias due to missing outcome data owing to high number of drop-outs. Hence, overall the risk of bias for the trial was high.”

3. The authors mentioned that “Given the small risk of RA with the exposure to statin, odds ratios, hazard ratios, and risk ratios were considered to have minimal differences and were combined in a single meta-analysis.”, which I think is incorrect. I have checked the included studies and found even both odds ratios (ORs) reported in the articles (for example, Schmidt [2013] and de Jong [2012]), there were significant differences in the way they calculated the ORs. This could also be one of reasons for the high heterogeneity. I would suggest using raw data (number of events) and pooled with the same ratio (OR or RR).

Response: We acknowledge this limitation of our review since some of the studies used OR while some used HR to report the association between statins and RA. This is now incorporated in the limitation section of the review. “Another factor to consider is that the included studies used different ratios, OR or hazard ratio or risk ratio to report the association between statin use and RA. Due to low prevalence of RA and limited number of studies in literature, we were forced to combine these ratios in a single meta-analysis. Given these ratios are not exactly the same, there could be some bias in our results.”

Since the number of studies in this review is limited and ratios differ, we are unable to separate studies based on the ratios used. Further, we stand by the statement made in the manuscript that in cases of low outcome rates (in this case RA), the different ratios are combined together in meta-analysis studies. This was also done in the prior systematic review of Myasoedova et al, quoted above by the reviewer, wherein they stated that: “The primary effect measures used in the studies were Odds Ratios (OR), Hazard Ratios (HR) and Relative Risks (RR). These effect measures were assumed to reasonably estimate the same association between statin use and RA occurrence given the low incidence of RA and thus were pooled together”

Secondly, we would like to stick with a meta-analysis of adjusted ratios as used in our meta-analysis and also the prior meta-analysis and not go in for raw data meta-analysis. Crude/raw data analysis is not adjusted for confounders and hence more prone to bias than combining different ratios together. 

4. As several of the included studies mentioned that intervention durations of less than or more than 1 year may affect the results. Subgroup analysis based on intervention duration is needed.

Response: As shown in Table 2, where we have collated all subgroup analysis reported by the included studies, only three studies reported data on the effect of duration of statin use and risk of RA. Peterson et al and de Jong 2018 classified as > or <1 year while de Jong 2012 classified the duration as 1-250 days, 251–600 days and ≥601 days. Given the scarce data and variation in classification, it is not possible to pool them into a subgroup analysis. Secondly, none of the remaining studies reported data on duration of consumption of statins to allow a subgroup analysis on this important variable. Lastly, follow-up data was also inconsistently reported by the studies which further doesn’t allow a subgroup analysis based on duration of follow-up. All these are already acknowledged in the limitation section of the discussion.

5. It is also important to conduct the leave-one-out analysis to determine whether the results could have been influenced by a single study.

Response: Sensitivity analysis is now added to the manuscript. 

.

---

## [Decision Letter · Decision Letter 1]

9 Jul 2024

Does the use of statins alter the risk of rheumatoid arthritis? A systematic review and meta-analysis

PONE-D-24-17747R1

Dear Dr. Qin,

We’re pleased to inform you that your manuscript has been judged scientifically suitable for publication and will be formally accepted for publication once it meets all outstanding technical requirements.

Kind regards,

Elena Olmastroni

Academic Editor

PLOS ONE

Additional Editor Comments (optional):

Reviewers' comments:

Reviewer's Responses to Questions

**Comments to the Author**

1. If the authors have adequately addressed your comments raised in a previous round of review and you feel that this manuscript is now acceptable for publication, you may indicate that here to bypass the “Comments to the Author” section, enter your conflict of interest statement in the “Confidential to Editor” section, and submit your "Accept" recommendation.

Reviewer #2: All comments have been addressed

2. Is the manuscript technically sound, and do the data support the conclusions?

Reviewer #2: Yes

3. Has the statistical analysis been performed appropriately and rigorously? 

Reviewer #2: Yes

4. Have the authors made all data underlying the findings in their manuscript fully available?

Reviewer #2: Yes

5. Is the manuscript presented in an intelligible fashion and written in standard English?

Reviewer #2: Yes

6. Review Comments to the Author

Reviewer #2: (No Response)

7. PLOS authors have the option to publish the peer review history of their article (what does this mean?). If published, this will include your full peer review and any attached files.

Reviewer #2: No

---

## [Editor Report · Acceptance letter]

12 Jul 2024

PONE-D-24-17747R1 

PLOS ONE

Dear Dr. Qin, 

I'm pleased to inform you that your manuscript has been deemed suitable for publication in PLOS ONE. Congratulations! Your manuscript is now being handed over to our production team.

Kind regards, 

on behalf of

Dr. Elena Olmastroni 

Academic Editor

PLOS ONE